# Influence of Beach Erosion during Wave Action in Designed Artificial Sandy Beach Using XBeach Model: Profiles and Shoreline

**Yingtao Zhou** [1,2,3,*], **Xi Feng** [1,2,4], **Maoyuan Liu** [4] **and Weiqun Wang** [3]

1   Hainan Key Laboratory of Marine Geological Resources and Environment, Haikou 570206, China
2   Key Laboratory of Marine Ecological Conservation and Restoration, Ministry of Natural Resources/Fujian Provincial Key Laboratory of Marine Ecological Conservation and Restoration, Xiamen 361005, China
3   Shanghai Urban Construction Design & Research Institute (Group) Co., Ltd., Shanghai 200125, China
4   College of Harbor, Coastal and Offshore Engineering, Hohai University, Nanjing 210098, China
*   Correspondence: zhouyingtao@sucdri.com

**Abstract:** Beach width is an important factor for tourists' comfort, and the backshore is a swash zone where sediment moves quickly. Artificial sandy beaches focus on beach width stability and evolution. This paper is based on an artificial beach project in Haikou Bay, where, in view of the existing conditions, a new type of beach profile that can protect beach berm and width without being eroded by large wave action. Numerical simulation based on XBeach model were conducted to predict the morphodynamical responses of the beach, including a diagnosis of the erosion spots under storm and normal wave events, respectively. Sediment fluxes along and across the shoreline under varied scenarios, dependent on profile width and backshore slope, were discussed. It was found that normal waves with lower heights and longer periods can induce stronger erosion than storm waves due to the landform of the inner-bay in Haikou Bay. Engineering and biological methods to reduce beach erosion during wave action were discussed. Biological methods such as green-plants-root-system can retain berm surface sediment without allowing it to be transported offshore by wave action. The design concept of this artificial beach project may inspire more beach design and protection projects in coastal zones.

**Keywords:** artificial beach; erosion; wave action; numerical simulation

## 1. Introduction

Beach width is an important factor for tourist comfort, while the backshore must protected from wave erosion induced by wave run-up and nearshore currents, which have strong effects, to protect the beach volume. With the construction of artificial beaches, a better understanding of the equilibrium mechanism of the beach surface is necessary for coastal engineering designs. With the construction of artificial beaches, a better understanding of the equilibrium mechanism of the beach surface is necessary for coastal engineering designs. Artificial beaches are usually designed into arc-shaped landscape. Therefore, existing curved embankments are often re-used for the development of the artificial beach. During normal wave events, such an arc-shaped beach profile favors morpho-dynamic balance, Extreme sea states such as storm tides and high water levels (HWL) has a profound influence on wave heights in estuaries and coastal areas, as well as sand movement [1]. The evolutionary processes of underwater sandbars and nearshore slopes are well understood, and some laws have been summarized [2]. Significant efforts have been made to reveal the erosion and restoration processes of beach profiles.. Multiple methods are combined to study the coastal alongshore variation including measurements [3] physical experiments [4] and numerical experiments [5–8]. Recently, the direct application of Radar (SAR) Satellite [9] and historical multispectral Landsat images analysis [10] in studying beach erosion and shoreline retreat has also become a popular trend.

Extreme wave actions process sufficient energy to erode and rebuild coastal regions, especially those with steep slopes or beaches with coarse-grained sediments, where the profiles are almost entirely controlled by waves rather than tide currents. Increasing evidence suggests global climate change will alter wind waves and swell waves, modifying the severity and frequency of episodic coastal flooding and morphological change,which in turn exacerbated the impacts of relative sea-level-rise [11,12]. It was found that onshore bedload transport by waves is the primary cause of the onshore migration of the shoreface substances. It was observed that the shoreface development increases local wave height and strengthens wave nonlinearity through its shallow water depth. The most intense wave-breaking dissipation has been found on the crest of the shoreface, while the distribution of wave energy dissipation rates is more uniform on the quasi-equilibrium profile than that on the initial profile [13].

Anthropogenic activities, such as development and reclamation of the estuary regions also contribute to the degradation of the coastal environment and have made it lose the resilience function of the buffer zone. Traditional seawalls and check dams are constructed mostly focusing on the protection of lives and assets, but a new problem arises when coastal hydraulic constructions change the dynamic balance of the geomorphological evolution in the natural coastal zone. These will impose negative effect on landscapes, ecosystems, biodiversity, and biological interactions in the long run [14]. Coastal areas will be eroded more than before with the sediment flux from upstream decreases or sediment unable to silt under the extreme sea-states, which happens in the Mississippi River Delta and Yangtze river deltas [14,15].

Fewer datasets are based on long-term series monitoring around the world [16–18], and long-term measurement plans are extremely expensive and will probably not assist us in building artificial beaches or protecting sediment profiles from erosion. In this paper, we designed artificial beach profiles after studying the nearshore wave and current conditions, then used the open-source numerical model XBeach to analyze its morphological behavior during storms and normal wave actions. In this article, three key questions need to be answered: (1) how to design stable artificial sandy beaches in profiles and shorelines based on the local geophysical characteristics? (2) whether the beach can keep stable under wave action from storm events or normal sea-states? (3) How to choose protective measures to keep shoreline balance, in terms of engineered and biological methods? This paper has been organized as follows. Section 2 provides brief descriptions of study sites. The numerical model and model setup are described in Section 3. Sections 4 and 5 demonstrate the project feasibility analysis, model result, and analysis. Discussion and conclusions are presented in Sections 6 and 7, respectively.

## 2. Study Site

The artificial beach has become one of the most popular attractions in the city of Haikou and has contributed profoundly to the city's tourism economy in recent years. The artificial beach is bounded by two newly built sand banks, which ensure the stability of water inside the bay. The sand banks are conducive to the maintenance of the artificial beach by shielding the offshore swells (Figure 1).

Haikou Bay is a semi-circular bay open to the north, from Baisha Cape in the east to Houhai in the west, across the sea from the Leizhou Peninsula in Guangdong Province, with a total coastline length of 20.5 km and water depth between 2 and 6 m. The artificial beach we built was approximately 916 m long (Figure 1a red line), based on the shallow water in front of the dam. The desilting and warping work on these sections was implemented in 2020 to clear and dredge the bed level to −2.8 m. The strip grooves on the backshore were approximately 100 m wide and 5~6 m deep.

There was a man-made island of 0.35 km$^2$ (Figure 1a), which was 1.5 km far away from that designed beach, and at the request of the central environmental protection and the Haikou government, the island was completely dismantled to −2.0 m.

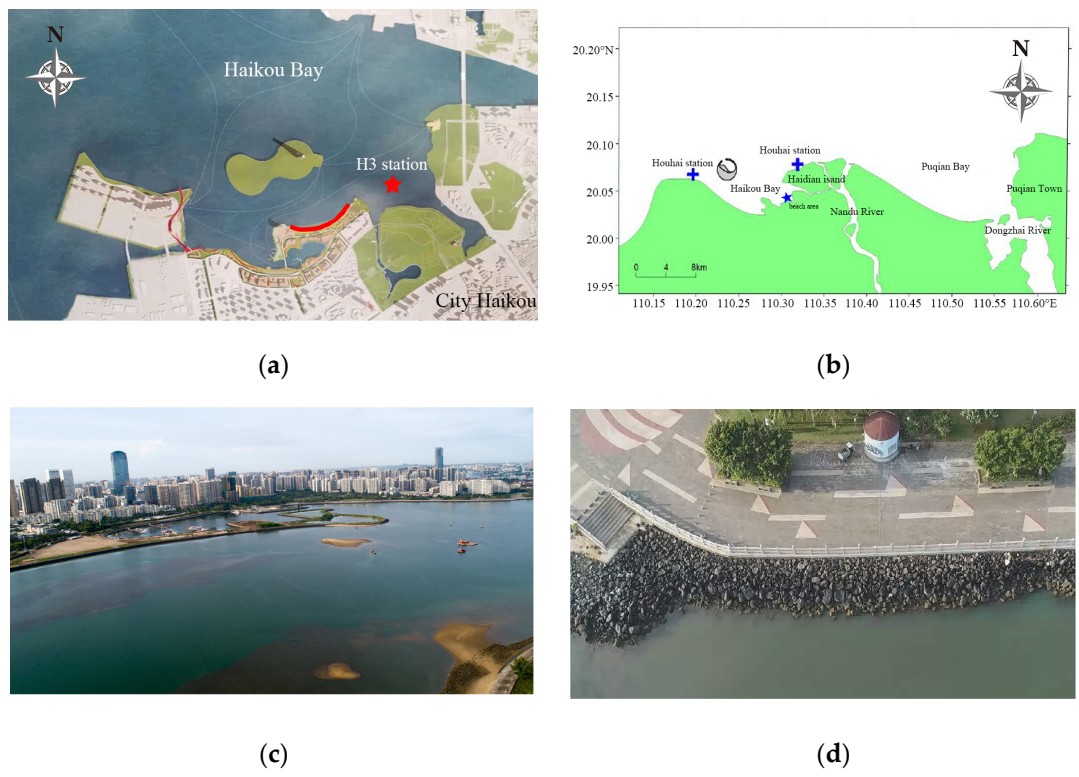

**Figure 1.** Haikou Bay sandy beach location. (**a**) Artificial beach location; (**b**) Wave buoy location; (**c**) Aerial of area before beach construction (I); (**d**) Aerial of area before beach construction (II).

Taking full account of the engineering planning, the depth and the nearshore hydrodynamic conditions, the artificial beach was designed with a shoreline length of 916 m. The width of the dry area (above MWL + 0.6 m) of approximately 30 m, the wet area (above ELWL + 0.6 m) of approximately 60 m, and protective stones were set the foot of the beach embankment was to reduce the loss of sand. The land boundary designed elevation was 2.20 m, and the dry beach outer boundary line was designed to have an elevation of 1.2 m. The artificial beach bottom layer within the existing breakwater was designed to fill the slag layer with the laying of a geotechnical anti-filter cloth layer. A throw pad layer of coarse sand, and a surface layer to fill 1 m thick using high-quality sand, with D50 = 0.2~0.6 mm. The total area of the sandy beach was expected to be 27,588 m$^2$. (Figure 2)

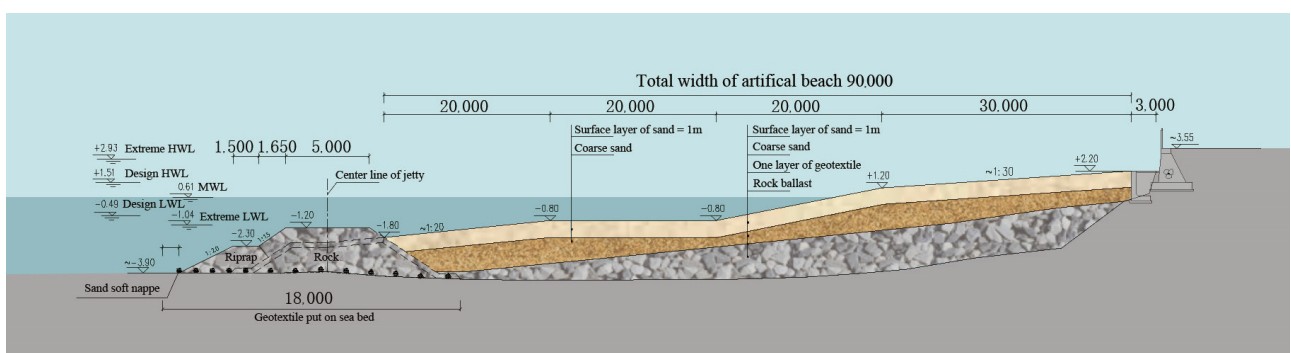

**Figure 2.** Rendering picture of designed sandy beach profile.

The high (HWL), mean (MWL), and low (LWL) water levels were 2.93 m, 0.61 m, and −0.49 m, respectively, based on the datum level in Haikou. The HWL and the LWL correspond to the mean high water and low water spring tides, respectively, after the analysis of the P-III type curve, and the results of the designed tide level are shown in

Figure 3. The wave conditions include both sea wind and swell waves, predominantly from the north (N-dir), NE (54%), and NNE (43%) directions. Most of the largest waves come from the north with heights exceeding 3.0 m, and with period with 6.0 s out in the bay. High energy (storm) wave conditions are generated between September and October by tropical cyclones, while waves are relatively small from October to June. The maximum wave heights induced by tropical cyclones are frequently less than 1 m in Haikou Bay.

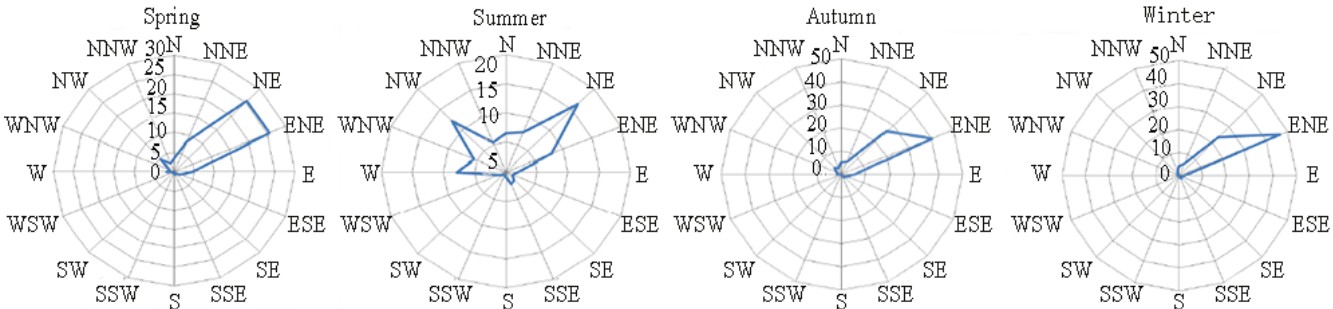

**Figure 3.** Wave rose chart in HOUHAI station.

## 3. Methods

### 3.1. Model Setup

XBeach is an open-source numerical model that was originally developed to simulate hydrodynamic and morphodynamic processes that impact sandy coasts with a domain size of kilometers and on the time scale of storms. Since then, the model has been applied to other types of coasts and purposes [19–22]. Using the surfbeat version of the model, the time-dependent, short wave action balance equation solves for the wave group envelope and is coupled with the non-linear shallow water equations to resolve mean currents and infragravity waves. Sediment transport is modeled using a depth-averaged advection-diffusion equation [23], and morphology change is calculated based on gradients in sediment transport at each time step. Dune erosion in Xbeach occurs directly from these transport gradients, as well as from avalanching, which is induced when a critical angle of repose is exceeded. Model formulations are described in detail in Roelvink [24,25] and are not repeated here. Xbeach has been extensively validated for dune erosion processes in numerous coastal settings [19–22].

The convective–diffusive equation to simulate sediment transport is as follows:

$$\frac{\partial hC}{\partial t} + \frac{\partial hC_u^E}{\partial x} + \frac{\partial hC_v^E}{\partial y} + \frac{\partial}{\partial x}\left[D_h h \frac{\partial C}{\partial x}\right] + \frac{\partial}{\partial y}\left[D_h h \frac{\partial C}{\partial y}\right] = \frac{hC_{eq} - hC}{T_s} \tag{1}$$

where $C$ is the suspended sediment concentration with non-dimensional, $h$ is bed level (m), $t$ is time (s), $u^E$ is Euler residual current (m/s), which is combined with wind-driven current, tidal current, infragravity waves (long period waves), offshore currents caused by nearshore breaking in the swash zone, and bottom reflux (time average velocity) (m/s). $u_a$ is non-linear time average velocity caused by short period waves (m/s), $u_a$, and $u_{rms}$ (practical water movements caused by wave) and represent the flow caused by short period waves. $\theta_m$ is mean wave direction, $D_h$ is the sediment diffusion coefficient, $C_{eq}$ is equilibrium concentration related to Euler residual currents and short period flows, and when the suspended sediment concentration is over this upper limit, sediment is deposited. $T_s$ is time with dependence on the settling velocity of silt particles and the bed level.

Based on the cross-shore designed profiles and shoreline, we carried out two-dimensional simulations along the total beach. The domain of the model with two coupling grids, the big computational domain extended from coastline to 2.0 km offshore, as shown in Figure 4a,b. With a grid size of 15 m × 15 m, smaller one is 5 m. The depth data in the grids were obtained by interpolation using the 5 m × 5 m resolution bathymetry data measured on December 2020 in Figure 4c.

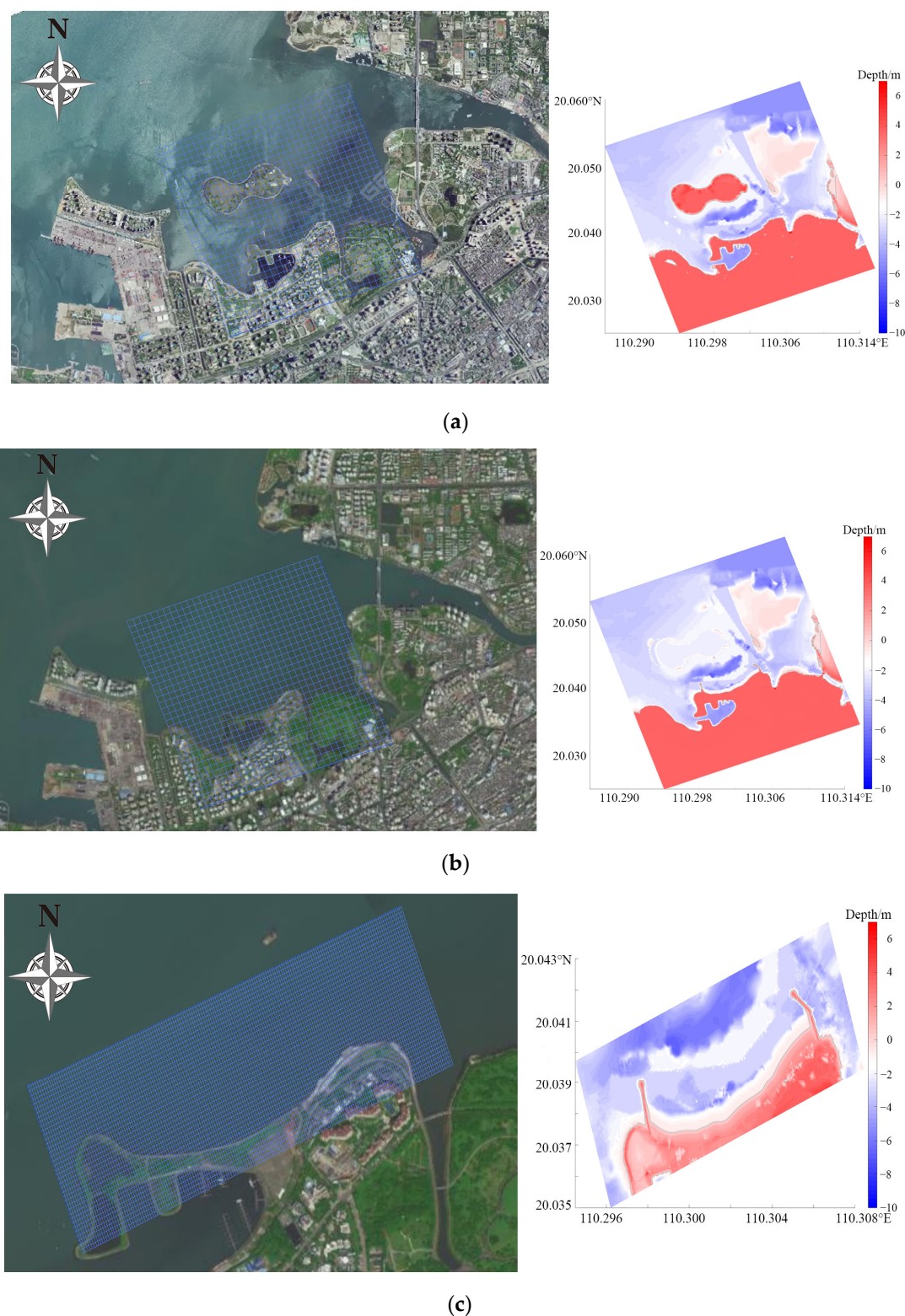

**Figure 4.** Model grid and depth in construction area. (**a**) model area with island; (**b**) Big model area without island; (**c**) Small model area.

The following simulation was based on the Surfbeat module in XBeach. The offshore boundary used the Neumann boundary condition and sediment transport was calculated with the Van Thiel-van Rijn equation as shown in Tables 1 and 2.

**Table 1.** Tidal characteristic value of Haikou (III) station.

| Site | MHWL/m | MLWL/m | HWL/m | MinT/m | MeanT/m | MaxT/m | DHWL/m | DLWL/m |
|---|---|---|---|---|---|---|---|---|
| Xiu Ying station | 1.17 | −0.04 | 2.93 | −0.86 | 1.21 | 3.79 | 1.51 | −0.49 |

**MHWL:** mean high water level; **MLWL:** mean low water level; **HWL:** highest water level; **LWL:** lowest water level; **Min T:** minimum tidal range; **Max T:** maximum tidal range; **DHWL:** design high water level; **DLWL:** design low water level.

**Table 2.** Design Water level in Haikou (III) station.

| Type | Mean Value | $p = 1\%$ | $p = 2\%$ | $p = 3.3\%$ | $p = 5\%$ | $p = 10\%$ | $p = 20\%$ |
|---|---|---|---|---|---|---|---|
| Annual maximum high tide level | 2.13 | 3.08 | 2.93 | 2.81 | 2.72 | 2.55 | 2.38 |
| Annual minimum low tide level | −0.84 | −1.07 | −1.04 | −1.01 | −0.99 | −0.94 | −0.90 |

The offshore forced conditions were provided by wave buoy data and several reports, which used the coupled wave, circulation, and sediment transport model: Delft3d [22], to simulate wave–current interaction in the Haikou Bay during the storms Haiou (T-HO), Weimaxun (T-W), and Haiyan (T-HY). The model results were modified with data measured at several measurement stations. In our Xbeach model, we used the hydrodynamic data extracted from their model results at the offshore boundary location (contains wave height Hs, wave period Tp, and wave direction Dir).

To illustrate the time period of storm and normal waves contributing different kinds of beach erosion, we superimposed modeled erosion rates for each of the three profiles we chose in the east, central, and west-most sections. The erosion rates were calculated by integrating the eroded volume per unit width over each hour. Table 3 summarizes the model parameters used in this study.

**Table 3.** Main parameter setting.

| Parameter | Description | Value | Reference Value |
|---|---|---|---|
| CFL | Maximum Courant–Friedrichs–Lewy number | 0.7 | 0.1~0.9 |
| morfac | Morphological acceleration factor | 5 | 0~1000 |
| bedfriccoef | Bed friction coefficient/s/m$^{1/3}$ | 0.02 | $3.5 \times 10^{-2}$~0.9 |
| wetslp | Critical avalanching slope under water | 0.3 | 0.1~1 |
| dryslp | Critical avalanching slope above water | 1.00 | 0.1~2 |
| D50 | D50 grain diameter first class of sediment | 0.4 | - |
| por | Porosity | 0.4 | 0.3~0.5 |
| reposeangle | Angle of internal friction/° | 30 | 0~45 |

*3.2. Wave Scenarios*

Multiple tests on storm and normal events were studied on designed beaches, with three typical storm (typhoon) events (No. 201415 T-HO; No. 201409 T-W; No. 201330 T-HY),which brought Haikou near-sea higher water levels and stronger waves causing damage and significant financial loss. The three storms had a duration of 39 h on average (defined as time when nearshore wind velocity > 3 m/s). The peak spectral wave period of the storms ranged from 1 to 1.5 s, and the direction of the waves varied throughout the storm as it passed northwest, ranging from 315 to 340 TN. The astronomical tidal range at the time of the storm was 2.80 m (mean spring tidal range in Haikou is 0.8 m).

From nearshore wave buoy data (B station and H station, Figure 1c) shows the waves during normal sea state are mostly driven by winds. Nearshore waves in the Haikou Bay area always contributed to N direction (N-dir) waves. Wind seas from NE and NNE directions, accounting for 54% and 43% in the H and B stations, respectively. During normal waves, almost all year round, waves are small and with periods between 3.0 s and 6.0 s. The largest waves come from N, with wave height over 3.0 m and wave period of 6.0 s at B station and wave height over 2.0 m with a period of 6.1 s at the H station. It can be

seen from the statistical data that the average monthly wave height in the project area is 0.2~0.7 m, which is below the designed high water level of the 50-year design period. The cumulative frequency at the revetment is one percent of the characteristic wave height H1%= 1.4 m, and Tp = 6.0 s/3.0 s. Wave data of stations B and H were used to describe wave conditions for the study domain. Waves were set with Hs = 1.4 m, and Tp = 3.0 s and 6.0 s respectively for normal sea states. Wave directions were selected as NW and N (Table 4). In addition, tidal forcing were considered simultaneously with wind forcing in the model.

**Table 4.** Model test in storms and normal events.

| Storm Number | Name | Starting Time (y/m/d) | Ending Time (y/m/d) | Fastest Wind Velocity (m/s) | Lowest Central Pressure (hPa) | Landing Location |
|---|---|---|---|---|---|---|
| 201415 | T-HO | 2014.09.11 | 2014.09.18 | 40 | 960 | WenChang |
| 201409 | T-W | 2014.07.11 | 2014.07.21 | 60 | 910 | WenChang |
| 201330 | T-HY | 2013.11.03 | 2013.11.12 | 42 | 955 | Vietnam |
| Model test Number | Hs (m) | Tp(s) | Model test Number | Hs (m) | Tp (s) | Dir (°) |
| Nom-1 | 1.4 | 6.0 | Nom-2 | 1.4 | 3.0 | NW |
| Nom-3 | 1.4 | 6.0 | Nom-4 | 1.4 | 3.0 | N |

## 4. Feasibility Analysis of Engineering Construction

Sampling data collected along the beach from the Xiu-Ying Habour to the north part of HaiDian island shows that the D50 is 0.01~0.67 mm. According to the site survey in Haikou Bay, the bedload near the project area is quite silty with the percentage of clay between 10% and 20%. The spatial distribution is discontinuous, and the sand content is approximately 0.01 kg/m$^3$. From the perspective of sediment distribution in the artificial beach area, this is mainly sandy silt with D50 < 0.2 mm. It could be observed that the west of the beach was mostly occupied by sandy silt. Due to the mud-cleaning project taken in Haikou Bay in 2020, the original mucky layer has been removed and sea-bed has been deeply dredged, which increase the nearshore wave energy and making the bed less prone to accretion. With the improved hydrodynamic environment and the gradual stabilization of the substrate, size of bedload sediment is likely to increase.

Due to the presence of silty clay nearshore, the movement of fine-grained sludge after the construction of the project was predicted and verified. The model used the H3 site (Figure 1a) with the measured sand content data (110°18′33.63″ E; 20°02′34.37″ N) for validation, which shows that the simulation results are compared well with measurements, and it can be seen from the simulation results (Figure 5c) that the silt movement trend in the area has not been significantly enhanced after the demolition of the island, and thus the beach surface will not become muddy.

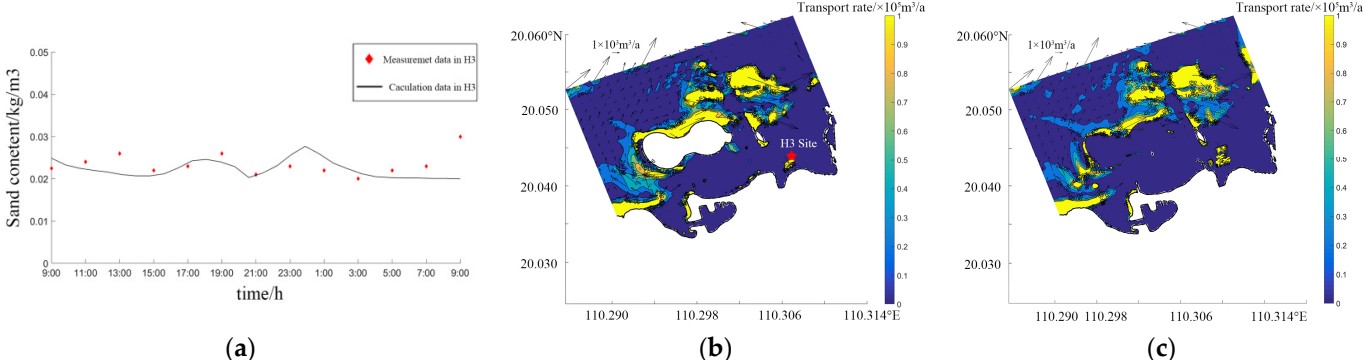

**Figure 5.** Sediment transport model validation and simulation. (**a**) Model validation with island. (**b**) Suspended sediment simulation with island. (**c**) Suspended sediment simulation without island.

## 5. Results

### 5.1. Profiles' Widths and Sediment Volume Change

The small area model utilizes the short-wave climb mode and can well simulate sediment movement above the MSL. Compared with Tp = 6.0 s and Tp = 3.0 s in normal wave actions (Figure 6(a1–a6)), we can see longer period waves cause greater beach erosion than shorter ones, and moving the sediment in larger scale with deeper erosion spots. With N and NW wave directions, the up limit of erosion spot caused by wave runup can reach 1.0 m above MWL. During N-dir waves, the erosion decreased from east to west due to the large shadow area of the west check dam and the beach shows alongshore variation. However, with NW-dir waves, the two sides of the check dams provide limited protection of beach volume, and the berm width displayed erosion in longshore. The nearshore volume (up volume) is nearly equal to backshore volume (down volume) in central while up volume is larger than down volume in west and opposite in east, which indicates that the sediment transport direction in the central and west parts of the beach was from east to west.

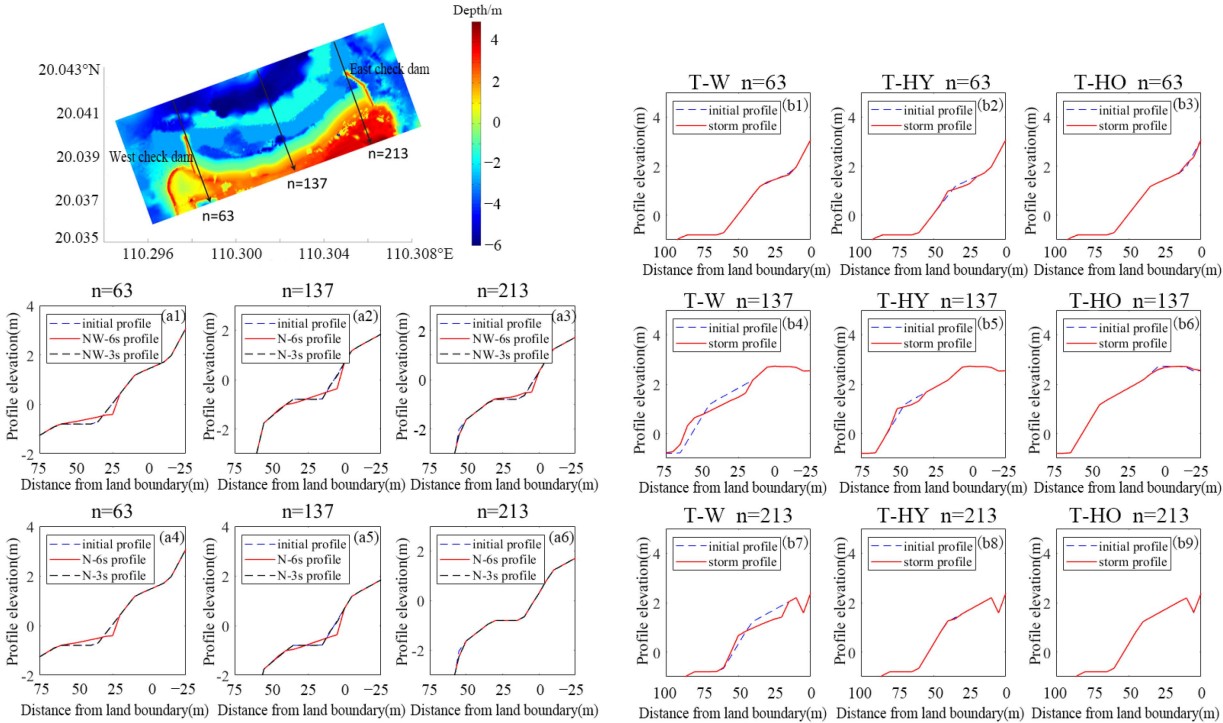

**Figure 6.** Comparison of profiles before and after storm event (**a1**–**a6**); before and after normal wave actions (**b1**–**b9**).

The evolution of three representative profiles (east, central and west of beach) from storm waves are shown in Figure 6(b1–b9). The width of beach is eroded and surface sediment move offshore to form sandbar under water. Wind velocity and storm duration time are main factors to induce profile erosion (Table 4). Stronger wind velocity and longer lasting time in T-W caused larger area of beach erosion than that during T-HO and T-HY events. Beach surface tends to be smooth after wave actions. T-W has caused more stronger erosion in central and west part of the beach than T-HO and T-HY, which swash zone is over 2.0 m above MWL, make more backshore erosion than T-HY and T-HO. The strongest profile volume change spot is located almost 700 m from east check dam, which illustrates great volume change between nearshore and backshore (elevation between −1.2 m and 2.0 m). Strong offshore and alongshore sediment transport in west part of the beach (Figure 7). Storm waves demonstrate no greater volume eroded than normal waves because

higher height wave can not keep energy without breaking, while lower energy wave with long period can transport to nearshore easily and beach will under continuous erosion.

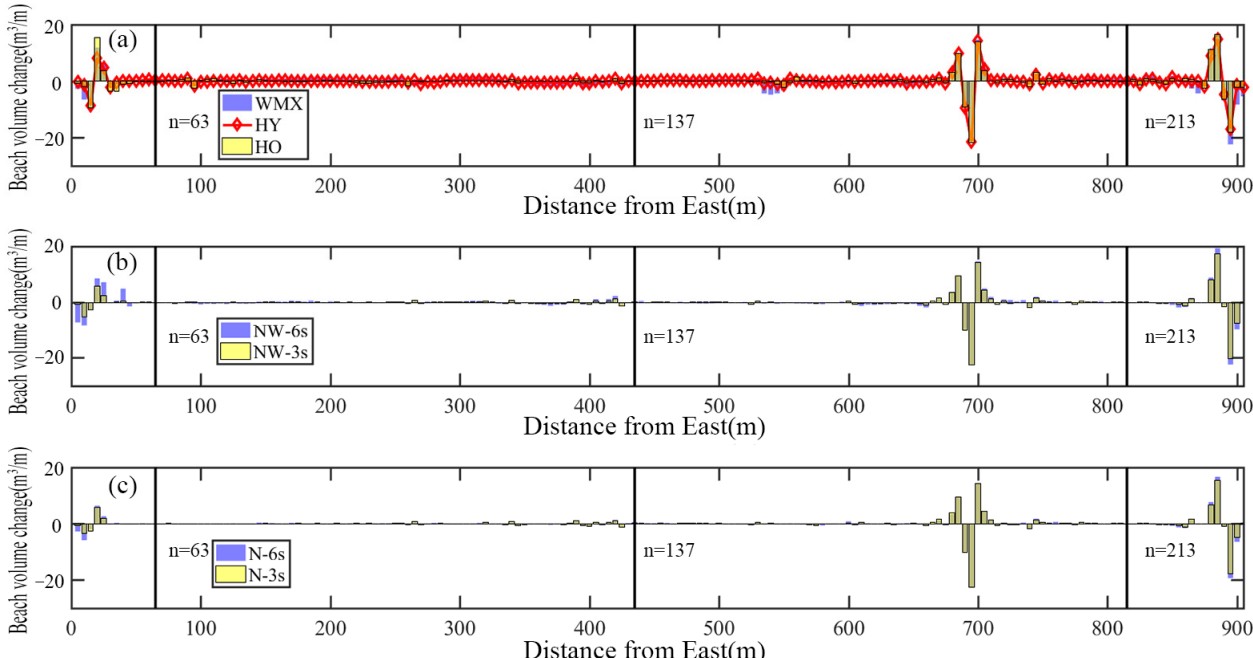

**Figure 7.** Beach volume change alongshore after storm and normal wave tests. (**a**). Beach volume change alongshore after storm wave events T-W, T-HY and T-HO. (**b**). Beach volume change alongshore after normal wave events Nom-1 and Nom-2. (**c**). Beach volume change alongshore after normal wave events Nom-3 and Nom-4.

During storms, the erosion of beach profiles is different from that of normal wave actions. Storm waves cause higher wave run-up and take away a large amount of sand in a few days, while recovery takes several months of normal wave actions. The width of the berm becomes narrower and the berm crest becomes lower during storms. The beach surface becomes smoother and sediment transport offshore from the berm, resulting in a reduction in beach width followed by sediment accretion below the LWL (middle profile). The east and west parts of the beach were found to be well protected on both sides by the check-dams.

### 5.2. Sediment Transport Alongshore and Offshore

Storm waves induced greater suspend sediment movement in beach berm between the MWL and the land boundary with concrete wall (0.61~2.2 m). Offshore is the main direction while alongshore transport is secondary. The T-W storm induced greater offshore sediment transport more than that during T-HY and T-HO, accounted for more than $5 \text{ m}^3/\text{d}$ in the central part of the beach storm. This is because wave propagates normal to the shoreline, the two-side check dams thus functions in beach protection.

Sediment transport in the nearshore area during normal wave events is much more different from storm waves, and it can be seen that offshore sandbar reduces the wave energy and contributes more sediment transport. From the simulations, we observed that due to the landform of the inner-bay in Haikou Bay, most high-energy waves break far offshore because of the sandbar and shallow water, but lower wave heights with long periods do not break, and reach the nearshore area, which have strong effect on suspend sediment transport. Thus, the Haikou Bay artificial beach should be well protected throughout year rather than just for storm wave events (Figure 8).

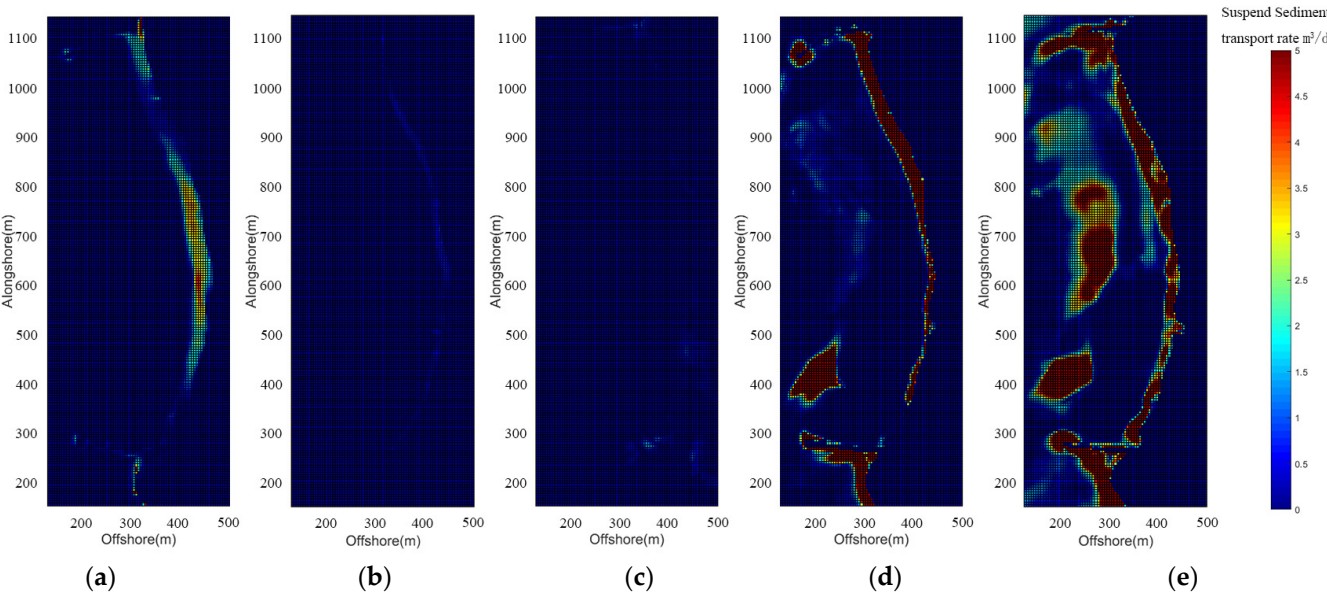

**Figure 8.** Suspend sediment transport due to wave actions in each model tests. (**a**) after T-W; (**b**) after T-HY; (**c**) after T-HO; (**d**) after N-6s normal waves; (**e**) after N-3s normal waves.

### 5.3. Shoreline Prediction

Study of beach shoreline variation during normal wave events not only provides more understanding of the wave-sediment-landform systems, but also the dynamics of sandy beaches. We used the empirical shoreline analysis tool MEPbay [26,27] to predict shoreline shape from long term wave actions. The tool presented in this paper only deals with the design shoreline and check dams. It was applicable to use wave angle and control line in two points and major physical parameters to get result of steady shoreline shapes. We are able to visually evaluate the stability of the beach by comparing the proximity of the existing shoreline with that predicted in static equilibrium.

Capes of the check dams and the central part of the beach was connected as a control line, then used NW and N as wave angle. This may be taken as the wave crest line, and is perpendicular to the incoming waves at the updrift diffraction point of the bay beach. Finally, we sketch the shoreline plane in static equilibrium. The east part of the beach was predicted to retreat due to the higher energy wave and less shadow zone in NW waves; while the west part of the beach was predicted in static equilibrium. In conclusion, the west part of the beach is predicted to become stable during normal waves, while the east part will become eroded, but the whole beach remains in dynamic-balance (Figure 9).

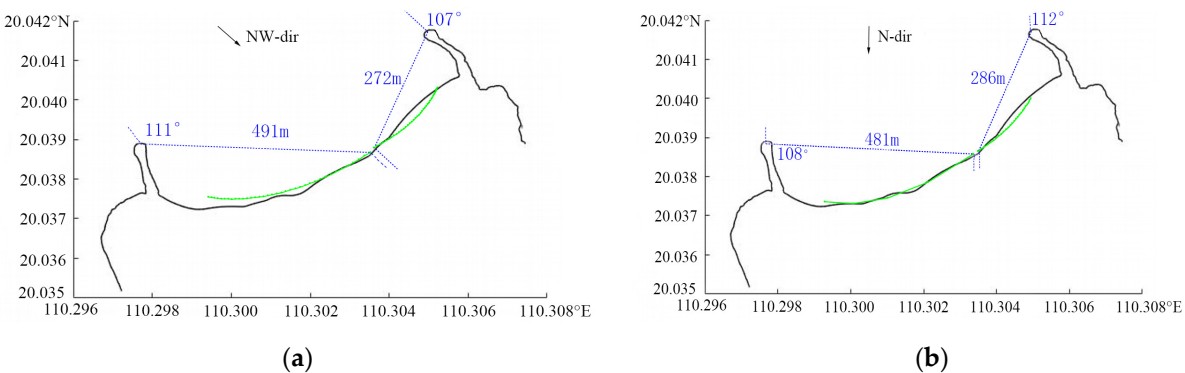

**Figure 9.** Prediction of Shoreline evolution after normal wave action. (**a**) Shoreline prediction in NW waves. (**b**) Shoreline prediction in N waves.

## 6. Discussion

### 6.1. Engineered Projects ("Grey Method")

Man-made constructions can increase/decrease wave energy, which induces beach erosion/deposition dramatically. Many unsuccessful engineering projects, such as the nearshore harbor construction, induced accretion of mud of the nearby beach with mud, which also shows us that changing dynamic conditions without prior assessment is dangerous and unbalances the ecosystem. It is necessary to keep the sediment without erosion and maintain the width of beach with using check dams and submerged breakwater. This constructions what we named "grey method" can make the wave breaking out of nearshore area and keep the sediment flux inside the beach area. But what we should also consider is lower wave energy do not mean better beach nourishment. Suspend sediment flux is inversely proportional to water depth, which means lower wave height will lead more suspend sediment moving to backshore and accretion on beach surface, thus the beach surface will become muddy.

### 6.2. Biological Methods ("Green Method")

Coastal resilience is a comprehensive indicator for coastal systems assessment. Sandy beaches are important areas for the turtles to lay their eggs, for birds migration and habitat, and for their predation. Coastal vegetation shows vertical richness from grass to shrub to trees, which provides a transition zone and has important environmental values. We emphasize coastal resilience not only enhance restoration after storm waves (i.e., the sand engine in Netherlands which was restored by natural dynamic conditions). Planting trees and grass can compensate for the loss of ecological niches and decrease abilities of sand movement which we called "soft" or "green" engineering, and we can see only minor erosion by storm waves due to the area of green plants (Figure 10) compared to concrete wall damaged without "greenery" (Figure 11). With these methods, we can build stronger and healthier coastal zone and achieve the goals of the "ocean decade plan".

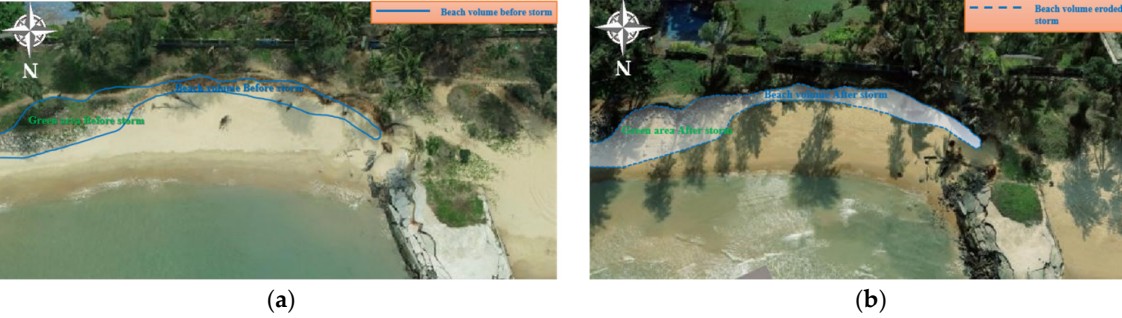

|  (a)  |  (b)  |
|---|---|

**Figure 10.** Beach berm with trees before (**a**) and after storm (**b**) in 2020 in Haikou Bay shows minor damage due to "green method" by storm waves.

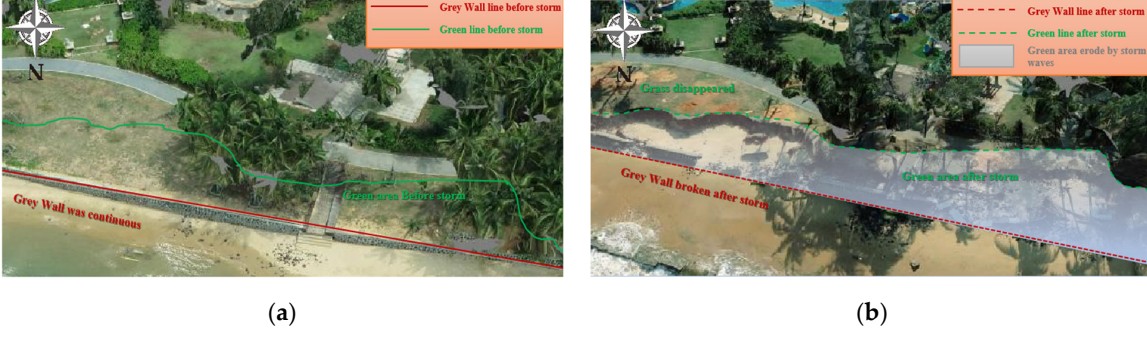

|  (a)  |  (b)  |
|---|---|

**Figure 11.** Beach berm with concert wall before (**a**) and after storm (**b**) in 2020 in Haikou Bay shows minor damage due to "grey method" by storm waves.

### 6.3. Offshore Infrastructure Induced Wave-Flow Field Change

From the "Results" section, we can see that normal wave action can induce more total beach erosion than storm waves. We tested normal wave events (Table 5) to illustrate how offshore infrastructure (the artificial island) caused wave refraction and flow field, although the offshore island was demolished in 2021 and the bedlevel was reduced to −2.0 m depth.

**Table 5.** Normal wave tests.

| Model Test Number | Topography | Hs (m) | Tp (s) | Model Test Number | Hs (m) | Tp (s) | Dir (°) | Explanation |
|---|---|---|---|---|---|---|---|---|
| 1 | With island | 1.4 | 6.0 | 4 | 1.4 | 3.0 | NW | |
| 2 | With island | 1.4 | 6.0 | 5 | 1.4 | 3.0 | N | |
| 3 | With island | 1.4 | 6.0 | 6 | 1.4 | 3.0 | NE | wave-tide coupling model |
| 7 | Without island | 1.4 | 6.0 | 10 | 1.4 | 3.0 | NW | |
| 8 | Without island | 1.4 | 6.0 | 11 | 1.4 | 3.0 | N | |
| 9 | Without island | 1.4 | 6.0 | 12 | 1.4 | 3.0 | NE | |

Note: **With island** means current topography (Figure 4a); **Without island** means island demolished, check dams and beach all completed (Figure 4b).

Due to the large area of shallow water, offshore waves will be dramatically reduced and well protected from the shadow zone behind the island (with island test) (Figure 12a–f). In wave current coupling model, the nearshore wave height becomes to 0.2~0.4 m. The wave from NE-dir has a closed angle with the beach shoreline, which contribute to wave height below 0.2 m nearshore. Without the island (Figure 12g–l), the artificial beach will face the open sea and wave-flow filed is improved (wave height and velocity will be greater). The predict wave height is 1.5 times higher than before and up to 0.4~0.6 m. The wave from NE-dir has a closed angle with the beach shoreline, which contribute to wave height below 0.2 m nearshore. Without the island (Figure 12g–l), the artificial beach will face the open sea and wave-flow filed is improved (wave height and velocity will be greater). The predict wave height is 1.5 times higher than before and up to 0.4~0.6 m. Wave height distribution is gradually reduced from east to west and the shadow zone remain on the sides of the two check dams. The N-dir wave is predict to shadowed in the central and western part of the beach, while NW-dir wave cannot be well defended because only a small shadow zone will provided by the check dams.

In the nearshore area, wave breaks and induces current velocity greater than 0.5 m/s. Under the conditions of NW-dir, N-dir and NE-dir waves, wave-induced currents reduce gradually from west to east. In the shadowed zone behind the island, the inshore velocity is below 0.1 m/s in all directions (Figure 13a–f). Nearshore current velocity increases after the island demolished but remains lower than 0.1 m/s, while the current at the check dams behind island area is predicted to become stronger. Without island (Figure 13g–l) velocity is predicted to be greater near the western check dam than the east, and thus contributing to significant erosion around the check dams, additional protection measurements should be considered around the western check dam.

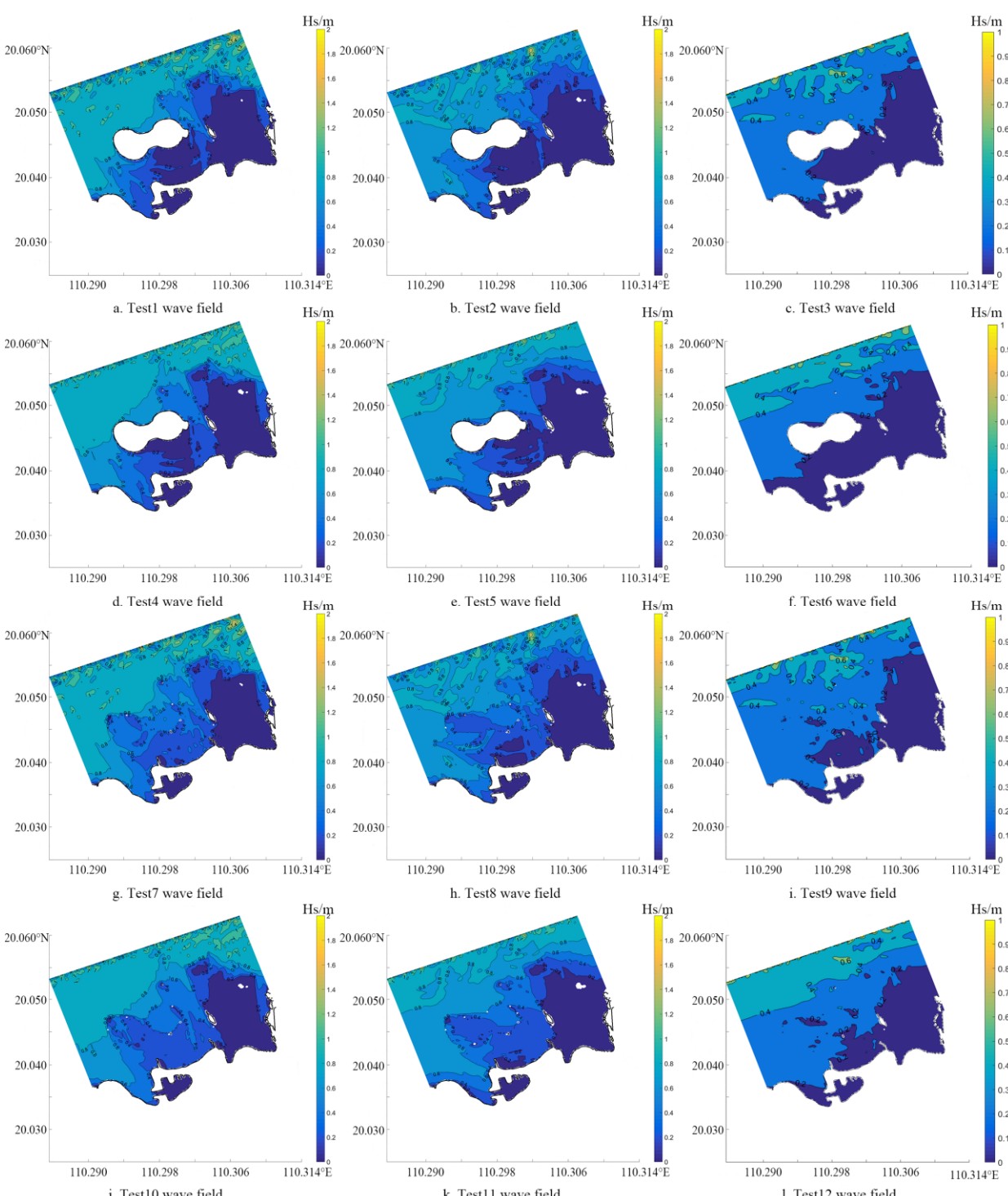

**Figure 12.** Wave filed in normal wave tests.



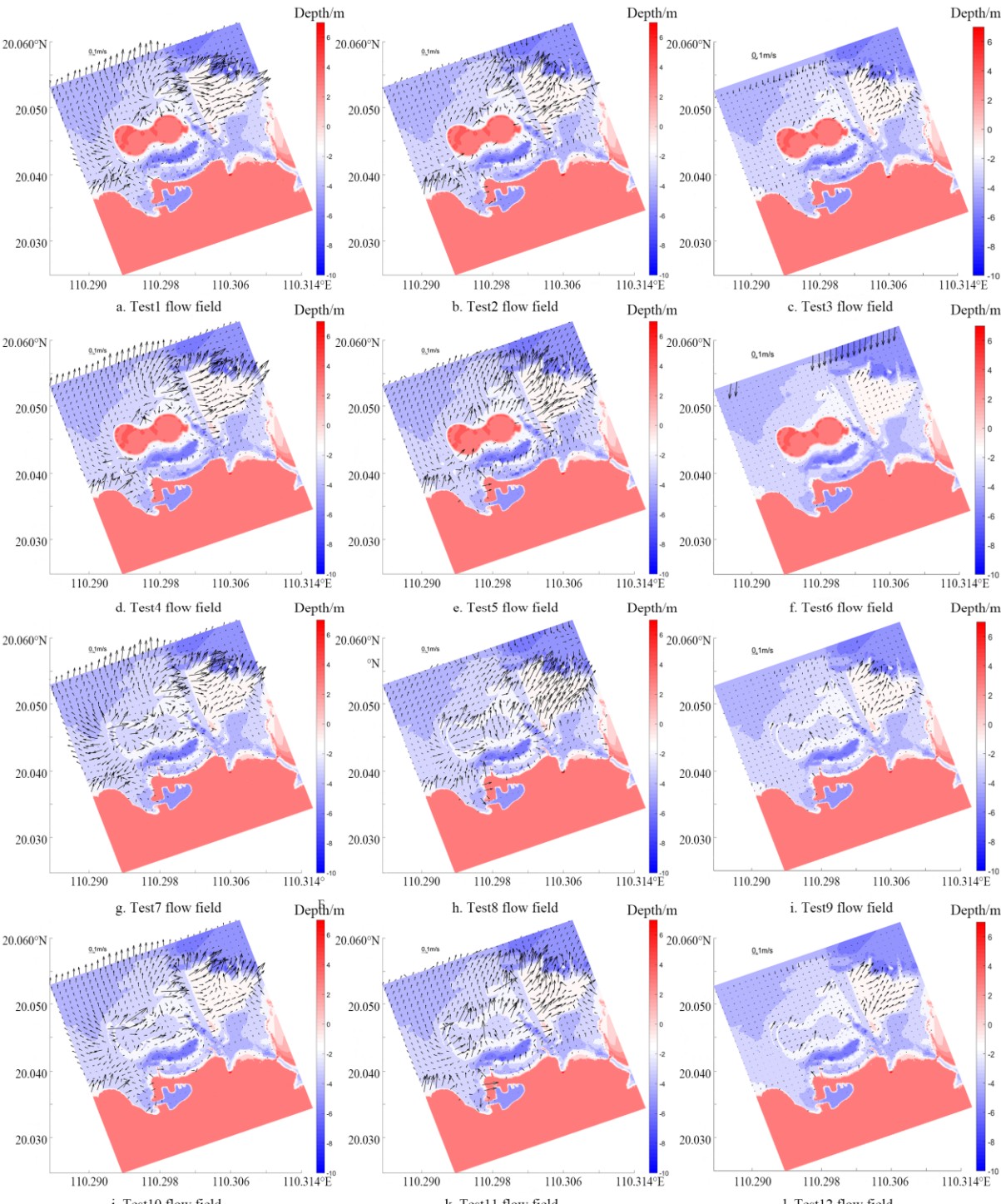

**Figure 13.** Flow filed in normal wave tests.

## 7. Conclusions

This study focuses on the beach profile evolution of an artificial sandy beach under both typical storm events and normal wave conditions, Main factors and hydrodynamics under varied sea-states associated with beach evolution were analyzed by using the Xbeach model. The main conclusions are as following:

(1) Normal waves induced greater erosion on the artificial beach than storm wave events.
(2) Waves with long wave period cause more erosion and accretion because they maintain long distance sediment transport without breaking.

(3) Sediment transport at the study site follows two main routes: offshore, and westward. It is argued that beach profiles will be eroded more in eastern portion of the beach than the western and central portion of the beach. The designed beach will maintain shoreline balance and remain in dynamic equilibrium during normal wave events.

(4) Offshore sandbars and ridges can protect the beach since wave energy can be largely reduced however they can enhance risks beach mudding and blackening in Haikou Bay.

Wave action is the main action to influence beach variation. It should be noticed that the resolved current field is vertically averaged by using this model. The physical processes are now resolved by semi-empirical functions in the XBeach model. In order to resolve more detailed physical processes in the surf zone where sediment entrainment and two phases flow could happen, more sophisticated 3D models are required.

**Author Contributions:** Conceptualization, Y.Z. and X.F.; methodology, Y.Z. and W.W.; software, Y.Z.; validation, Y.Z. and M.L.; writing—original draft preparation, Y.Z.; writing—review and editing, Y.Z. and M.L.; figure, M.L.; supervision, X.F. All authors have read and agreed to the published version of the manuscript.

**Funding:** This work was financially supported by the Fund of Hainan Key Laboratory of Marine Geological Resources and Environment (22-HNHYDZZYHJKF028) and The project was supported by the Fund of Key Laboratory of Marine Ecological Conservation and Restoration, Ministry of Natural Resources/Fujian Provincial Key Laboratory of Marine Ecological Conservation and Restoration, (EPR2023009).

**Institutional Review Board Statement:** Not applicable.

**Informed Consent Statement:** Not applicable.

**Data Availability Statement:** Data will be made available on request.

**Acknowledgments:** The authors are very much grateful to the four anonymous reviewers for their constructive comments that significantly improved the quality of this paper.

**Conflicts of Interest:** The authors declare no conflict of interest. The funders had no role in the design of the study; in the collection, analyses, or interpretation of data; in the writing of the manuscript, or in the decision to publish the results.

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
