# Peer review of "Influence of Beach Erosion during Wave Action in Designed Artificial Sandy Beach Using XBeach Model: Profiles and Shoreline"

_jmse, doi:10.3390/jmse11050984_

Round 1

Reviewer 1 Report

The work is well structured and highlights the current situation, means of intervention and advanced analysis methods. Special congratulations for the biological method of ensuring the stability of Haikou beach.

Author Response

The work is well structured and highlights the current situation, means of intervention and advanced analysis methods. Special congratulations for the biological method of ensuring the stability of Haikou beach.

Answer:Thank you very much for your comments. All the figures have been modified and  so for the English writing.

Reviewer 2 Report

This paper will accept if the following amendment done by the author:

·       The objective and motivation should be discussed in the introduction section, and tell people what advantages your paper involves which still need to be added.

·       The abstract is wordy and not informative. The structure of the abstract needs revision. Revise the abstract to provide

(i)             the significance of the study,

(ii)           (ii) the aim of the study,

(iii)         (iii) the research methodology,

(iv)          (iv) the major conclusion of the study.

·       English language needs some improvement throughout the paper. Example in the Abstract “Moreover, which found that present results found to be the present results”

·       On what basis the parametric values are chosen? Do they correspond to a specific physical condition? Please justify?

·       Authors should related to their work with real life application which specified?

·       The literature review suffers from significant self-citations and is not comprehensive given the many effects and should be improved by considering the following related references: Effects of MHD on modified nanofluid model with variable viscosity in a porous medium, Effects of MHD on modified nanofluid model with variable viscosity in a porous medium, Transportation of slip effects on nanomaterial micropolar fluid flow over exponentially stretching, Influence of Lorentz force and induced magnetic field effects on Casson micropolar nanofluid flow over a permeable curved stretching/shrinking surface under the stagnation region.

·       The introduction is very poorly written like as ‘‘check the grammar?

·       Why the author study this model? Author also add the nomenclature?

Author Response

This paper will accept if the following amendment done by the author:

The objective and motivation should be discussed in the introduction section, and tell people what advantages your paper involves which still need to be added.

Answer:Thank you for your advice and suggestions. We have modified introduction section as follows (Line76-85):

“In this article three key questions need to be answered: (1) how to design stable artificial sandy beaches based on project location and characteristics? (2) how to make sure the beach is well protected from wave action and analyze different characteristics of storm and normal wave events? (3) Compare these with engineered and biological methods and discuss how to keep shoreline balance? This paper has been organized as follows. Section 2 provides a brief description of study sites. The numerical model and model setup are described in Section 3. Sections 4 and 5 demonstrate the project feasibility analysis, model result and analysis. Discussion and conclusions are presented in Sections 6 and 7, respectively. ”

The abstract is wordy and not informative. The structure of the abstract needs revision. Revise the abstract to provide

(i)the significance of the study,

(ii)the aim of the study,

(iii)the research methodology,

(iv)the major conclusion of the study.

Answer:Thank you for your advice and suggestions, the modified abstract is shown as follows(Line11-25):

Beach width is an important factor for tourists comfort, while the back shore is a swash zone where sediment moves quickly. Artificial sandy beaches focus on beach width stability and evolution. This paper is based on an artificial beach project in Haikou Bay, where, in view of the existing conditions, a new type of beach profile that can protect beach berm and width without being eroded by offshore extremely high waves was put forward. This example may assist those institutions that want to establish artificial beaches or island coastal zones. The anti-erosion sandy beach system we designed maintained balance during wave action. Numerical simulation based on XBeach model is used to predict morphodynamical responses of the beach including diagnosis of the erosion spots, under storm and normal wave events, respectively. Sediment fluxes along and across the shoreline under varied scenarios, dependent on profile width and back shore slope, were discussed. It is found that normal waves with lower heights and shorter periods can induced stronger erosion than storm waves due to the inner-bay landform in Haikou Bay. Engineering and biological methods to reduce beach erosion during wave action were discussed. Biological methods such as green-plant-root-system can retain berm surface sediment without allowing it to be transported offshore by wave action.

  • English language needs some improvement throughout the paper. Example in the Abstract “Moreover, which found that present results found to be the present results”

Answer:Thank you for your advice and suggestions, English writing have been improved through professional English agencies.

  • On what basis the parametric values are chosen? Do they correspond to a specific physical condition? Please justify?

Answer:Due to this artificial beach is new project that there is no measured data before, we use H3 suspended sediment measured site near our beach area(Figure1) to calibrate the model. Then we use designed D50 in this project to run this model.

Table. 3  Main parameter setting

Parameter

Description

Value

Reference value

CFL

morfac

bedfriccoef

wetslp

dryslp

D50

por

Maximum courant-friedrichs-lewy number

Morphological acceleration factor

Bed friction coefficient /s/m1/3  

Critical avalanching slope under water

Critical avalanching slope above water

D50 grain diameter first class of sediment

Porosity

0.7

5

0.02

0.3

1.00

0.4

0.4

0.1~0.9

0~1 000

3.5´10-2 ~0.9

0.1~1

0.1~2

-

0.3~0.5

reposeangle

Angle of internal friction /°

30

0~45

  • Authors should related to their work with real life application which specified?

Answer: We designed artificial beach based on local conditions, which can refer to engineer construction, we also advised our abstract and introduction to make it more related to real engineering application.

  • The literature review suffers from significant self-citations and is not comprehensive given the many effects and should be improved by considering the following related references: Effects of MHD on modified nanofluid model with variable viscosity in a porous medium, Effects of MHD on modified nanofluid model with variable viscosity in a porous medium, Transportation of slip effects on nanomaterial micropolar fluid flow over exponentially stretching, Influence of Lorentz force and induced magnetic field effects on Casson micropolar nanofluid flow over a permeable curved stretching/shrinking surface under the stagnation region.

Answer: Thank you very much for suggestion. The references list has been updated based on the revision of the entire article. The most relevant studies with focus on the nearshore processes and storm beach evolution were included. 

  • The introduction is very poorly written like as ‘‘check the grammar?

Answer: Thank you very much for suggestion. The English writing has been modified in the revision.

  • Why the author study this model? Author also add the nomenclature?

Answer: The XBeach model is an open-source numerical model which is originally developed by Delft University of Technology (https://oss.deltares.nl/web/xbeach/). It was created specifically for resolving nearshore issues, including wave propagation, sediment transport and morphological changes. It has been successfully applied in many case studies focused on beach evolution.

Reviewer 3 Report

The authors try to discuss the influence of beach erosion during wave action in designed artificial sandy beaches using XBeach model: profiles and shoreline. The topic is interesting and suitable for publication after following concerns are addressed.

1.       Line-34-35 please check the grammar.

2.       Line 68-69 please modify the sentence, please avoid qualitative sentences.

3.       Please improve the resolution of fig.3

4.       Authors should improve the conclusion with bullet points. They should also discuss the application and limitation of their model.

5.       Abstract should be amended with some conclusive result at the end.

6.       English should be improved throughout the manuscript and redundancy should be modified.

7.       More recent literatures should be cited, especially those from JMSE-MDPI

8.       Fig.10, please increase the font size of the scale in the axes.

9.       There to Fig.10 in the manuscript, please correct figure no.

10.   In fig.10 of, please mark the beach's erosion-prone area.

Author Response

1.Line-34-35 please check the grammar.

Answer: Thank you very much for pointing out the detailed writing issue. Line34-35 have been modified as following:

Significant efforts have been made to reveal the erosion and restoration processes of beach profiles before and after storms through measurements[6], physical[7] or numerical experiments [8-10]. That reveal the coastal alongshore variation[11]. Recently, direct application of Radar (SAR) Satellite[12] and historical multispectral landsat images analysis[13] on studying beach erosion and shoreline retreat has also become a popular trend.

2.Line 68-69 please modify the sentence, please avoid qualitative sentences.

Answer: We have already modified the sentence, it is shown as follows(Line89-93):

The artificial beach has become one of the most popular attractions for city of Haikou and  contributed profoundly to the city’s tourism economy in recent years. The artificial beach is bounded by two newly-built sand banks, which ensure the stability of water inside the bay. The sand banks are conducive to the maintenance of the artificial beach by shielding the offshore swells (Fig. 1).

3.Please improve the resolution of fig.3

Answer: The figure resolution has been improved.

4.Authors should improve the conclusion with bullet points. They should also discuss the application and limitation of their model.

Answer: Thank you for your advice and suggestions and the last section has been revised including the limitation of this model, which is shown as follows (Line391-396):

It should be noticed that the resolved current field is vertically averaged by using this model. In order to resolve more detailed physical processes in the surf zone like sediment entrainment, two phases flow, more sophisticated 3D model would be required. The physical processes are now resolved by semi-empirical functions in Xbeach, whilst functions efficiently and successfully in practice.

5.Abstract should be amended with some conclusive result at the end.

Answer: Thank you for your advice and suggestions, the abstract section has been revised.

6.English should be improved throughout the manuscript and redundancy should be modified.

Answer: English writing has been improved.

7.More recent literatures should be cited, especially those from JMSE-MDPI

Answer: We have referred more papers from JMSE, most of them were added in the introduction part, such as:

[12]Zollini, S.; Dominici, D.; Alicandro, M.; Cuevas-González, M.; Angelats, E.; Ribas, F.; Simarro, G. New Methodology for Shoreline Extraction Using Optical and Radar (SAR) Satellite Imagery.  . Mar. Sci. Eng. 2023, 11, 627. https://doi.org/10.3390/jmse11030627

[13]Vallarino Castillo, R.; Negro Valdecantos, V.; Moreno Blasco, L. Shoreline Change Analysis Using Historical Multispectral Landsat Images of the Pacific Coast of Panama. J. Mar. Sci. Eng. 2022, 10, 1801. https://doi.org/10.3390/jmse10121801

[14]Li, Yuan, et al. "Wave Dissipation and Sediment Transport Patterns during Shoreface Nourishment towards Equilibrium."Journal of Marine Science and Engineering 9.5 (2021): 535.

[19]Cristaudo, D.; Gross, B.M.; Puleo, J.A. Momentum Balance Analysis of Spherical Objects and Long-Term Field Observations of Unexploded Ordnance (UXO) in the Swash Zone. J. Mar. Sci. Eng. 2023, 11, 79. https://doi.org/10.3390/jmse11010079

[20]Kuang, C.; Ma, Y.; Han, X.; Pan, S.; Zhu, L. Experimental Observation on Beach Evolution Process with Presence of Artificial Submerged Sand Bar and Reef. J. Mar. Sci. Eng. 2020, 8, 1019. https://doi.org/10.3390/jmse8121019

[21]Yang, Z.; Yang, Z.; Deng, Z.; Chen, Y.; Yang, B.; Hou, Y.; Deng, Z.; Tong, M. Multi-Timescale Analysis of the Evolution of Sandy Coastline: A Case Study in South China. J. Mar. Sci. Eng. 2022, 10, 1609. https://doi.org/10.3390/jmse10111609

8.Fig.10, please increase the font size of the scale in the axes.

Answer: We have already modified all figures to make them more easier and clear to read

9.There to Fig.10 in the manuscript, please correct figure no.

Answer: We have already correct the figure number.

10.In fig.10 of, please mark the beach's erosion-prone area.

Answer: We have already marked the erosion-prone area in the beach.

Reviewer 4 Report

This paper is good but the quality of all figures is not good. Please revise it properly. 

Author Response

This paper is good but the quality of all figures is not good. Please revise it properly.

Answer: Thank you for your advice and suggestions. All the figures are refreshed and the English writing has been modified.

Reviewer 5 Report

Paper is well written. Interesting applications. Proposed for possible publications. 

Author Response

Paper is well written. Interesting applications. Proposed for possible publications.

Answer: Thank you for your advice and suggestions. All the figures are refreshed and the English writing has been modified.

Round 2

Reviewer 2 Report

I accepted as present form. 

Reviewer 4 Report

The authors tried to improve the quality of the figures as suggested in the previous review. However, they failed completely and there is no or very less improvement in the figures' quality. In most of the figures, the text is not visible and even the surface color plots are blurred. I strongly recommend rejecting this paper.